# Immunomodulatory Activities of Selected Essential Oils

**DOI:** 10.3390/biom10081139

**Published:** 2020-08-03

**Authors:** Georg Sandner, Mara Heckmann, Julian Weghuber

**Affiliations:** 1Center of Excellence Food Technology and Nutrition, University of Applied Sciences Upper Austria, Stelzhamerstraße 23, 4600 Wels, Austria; georg.sandner@fh-wels.at (G.S.); Mara.Heckmann@students.fh-wels.at (M.H.); 2FFoQSI GmbH-Austrian Competence Centre for Feed and Food Quality, Safety and Innovation, Technopark 1C, 3430 Tulln, Austria

**Keywords:** essential oils, immunomodulatory, eucalyptus, clove, tea tree, lavender

## Abstract

Recently, the application of herbal medicine for the prevention and treatment of diseases has gained increasing attention. Essential oils (EOs) are generally known to exert various pharmacological effects, such as antiallergic, anticancer, anti-inflammatory, and immunomodulatory effects. Current literature involving in vitro and in vivo studies indicates the potential of various herbal essential oils as suitable immunomodulators for the alternative treatment of infectious or immune diseases. This review highlights the cellular effects induced by EOs, as well as the molecular impacts of EOs on cytokines, immunoglobulins, or regulatory pathways. The results reviewed in this article revealed a significant reduction in relevant proinflammatory cytokines, as well as induction of anti-inflammatory markers. Remarkably, very little clinical study data involving the immunomodulatory effects of EOs are available. Furthermore, several studies led to contradictory results, emphasizing the need for a multiapproach system to better characterize EOs. While immunomodulatory effects were reported, the toxic potential of EOs must be clearly considered in order to secure future applications.

## 1. Introduction—The Immune System and Herbal Medicine

Recently, there has been growing interest in investigating the immunomodulatory activities of essential oils (EOs) and their individual components. EOs are highly concentrated natural oils derived from plants that consist of aromatic, volatile, secondary plant metabolites. EOs are mostly extracted by steam distillation and exhibit a very intense odor. The main compounds in EOs represent mono- and sesquiterpenes and several oxygenated derivatives. Other substances found in the oils include alcohols, aliphatic aldehydes, and esters. Usually, EOs comprise two or three main constituents that determine their biological activities and chemical properties [1,2]. For example, clove essential oil consists of 70–76% eugenol, a phenolic compound responsible for its antimicrobial and antioxidant activities [3,4]. However, the bioactivity of EOs is influenced by the full composition of the oil, including minor components and possible synergistic, additive, as well as antagonistic effects [5]. Different bioactivity profiles within the same EO as a result of variations in the chemical composition of the oil have been reported [6]. Factors, such as the environmental conditions (temperature, light, location), physiology of the plant (plant age and plant parts), and genetic aspects, highly influence the chemical composition and thus biological activity of these oils [7]. Novel chemometric approaches could predict the bioactivity of multicomponent substances more reliably [8,9]. The applications of EOs are immensely diverse and highly dependent on the plant source. EOs are widely used in aromatherapy, cosmetics, the food industry (flavoring and preservative agents), and the pharmaceutical industry (antimicrobial and analgesic agents). The interest in further applications of EOs is steadily increasing. Their immunomodulatory activities are mediated through multiple mechanisms: EOs have been found to stimulate the immune system by increasing the amount of circulating lymphocytes and enhancing their phagocytic activity, thus improving bacterial clearance [10,11]. EOs have also been shown to suppress responses involved in inflammation and decrease cytokine production by interfering with key mediators of inflammatory pathways [1,2,3].

The mammalian immune system is functionally divided into two main compartments: the innate and adaptive immune systems. The innate immune system initiates primary defense reactions and facilitates the elimination or containment of infectious agents by inducing an inflammatory response. Effector cells of the innate immune system include natural killer cells, mast cells, basophils, phagocytic macrophages, monocytes, dendritic cells, neutrophils, and eosinophils. These cells are involved in phagocytosis, cytokine production, antigen presentation, and the release of inflammatory mediators. Invading pathogens are detected via pattern recognition receptors that bind to characteristic foreign structures, such as unmethylated DNA or the bacterial cell wall component, lipopolysaccharide (LPS). This binding activates the cells, triggering phagocytosis of the pathogen and inducing inflammatory mechanisms, including the activation and recruitment of additional effector cells to the site of infection through the secretion of cytokines and chemokines (Figure 1) [4].

The activation of these nonspecific innate effector mechanisms is critical for the stimulation of the adaptive immune response, a delayed but highly specific defense against the respective infectious agent [10]. The adaptive immune system is characterized by the clonal selection of lymphocytes that respond to a specific antigen and mainly involves antibody-producing B cells and T cells, particularly CD4^+^ T helper cells and CD8^+^ cytotoxic T cells. Naïve CD4^+^ T helper (Th) cells differentiate into different subtypes depending on the influencing cytokines. Th1 cells induce inflammatory responses and support the immune system in fighting intracellular pathogens, whereas Th2 cells primarily aid the differentiation of B cells into antibody-producing plasma cells. While extracellular pathogens are contained through humoral, antibody-mediated immune responses, intracellular pathogens are eliminated in the course of cell-mediated immunity, which is a delayed response mainly involving T cells, macrophages and natural killer cells.

However, since activation of the immune response is linked to tissue damage or destruction, inflammatory reactions are regulated by numerous mechanisms and are usually immediately terminated once the cause of the inflammation has been eliminated [12]. A loss of regulation of inflammatory responses may lead to the development of chronic inflammatory diseases, such as asthma, atherosclerosis, and inflammatory bowel disease [13]. Furthermore, immunodeficiencies can occur when certain components of the innate and adaptive immune system fail due to inherited genetic defects or damage from external factors, such as medication or nutrition [4].

Immunomodulation refers to any processes that alter the immune system either by enhancing (immunostimulation) or suppressing its function. While immunostimulation occurs through the activation of inactive components of the immune system or the augmentation of their activity, immunosuppression describes a reduction of the efficacy of immune responses. This reduction can be beneficial when it assists in diminishing inflammatory and autoimmune responses [14].

Herbal medicine and its applications have gained increasing attention regarding the prevention and treatment of diseases in recent years. Several plants and their extracts have been successfully found to affect T cells and cytokine and antibody production at the cellular and molecular levels [15]. For example, cinnamon bark oil and clove bud oil were shown to increase villi height in the *duodenum*, *jejunum,* and *ileum* of broilers. Antibody titers against Newcastle disease virus were also increased compared to those of control groups. Thus, EOs have been identified as alternatives to antibiotic growth promotors in broilers. Tea tree oil was applied to weaned piglets and improved intestinal mucosal immunity by increasing the interleukin levels IL-2 and IL-10, as well as interferon-γ (IFNγ), in the *jejunum* and *ileum*. Additionally, the villus length was improved by this treatment. The data revealed that tea tree oil was more effective than standard antibiotics. Tea tree oil was shown to reduce diarrhea and improve growth performance in piglets [16]. Also, ginseng extracts and their EOs were shown to induce immunostimulatory effects, for example by increasing tumor necrosis factor (TNFα) and IFNγ levels or enhancing phagocytic activity [17,18,19].

Therefore, the promising immunomodulatory effects of plant essential oils could be utilized for alternative treatments. Additionally, natural compounds often show better patient compliance with fewer side effects than standard pharmaceutical drugs. This review focuses on the immunomodulatory responses of four common essential oils (eucalyptus, clove, tea tree, and lavender) that showed the most promising effects according to literature.

## 2. Methods

Literature research was conducted using PubMed, ScienceDirect, and Google Scholar databases and the main searching terms “essential oils”, “immunomodulation”, “immunostimulation”, “immunosuppression”, and “inflammation”. Prominent essential oils affecting the immune response (mainly documented in eucalyptus, clove, tea tree, and lavender) were then added to the search history resulting in specific publications after the initial literature research. In this review, we focused on recent articles and reviews (2015–2020). Articles presenting interesting findings regarding the four selected essential oils and immune response were chosen for this review. A list of abbreviations is shown in the Appendix A. Results are presented and discussed in the following sections.

## 3. Eucalyptus Essential Oil

The genus *Eucalyptus* belongs to the *Myrtaceae* family and comprises approximately 800 species of flowering trees and shrubs that are native to Australia but have been cultivated around the globe [20]. *Eucalyptus* contains large amounts of volatile compounds (e.g., 1,8-cineole, α-pinene, citronellal and linalool; a summary of the main components is shown in Figure 2 and Table 1), which are mostly found in the leaves of the plant [21]. *Eucalyptus* essential oil (EEO) has a long history of medicinal use in the treatment of cough, cold, influenza, and other respiratory infections. The main component is 1,8-cineole (eucalyptol), a monoterpene responsible for its pungent, sharp scent and its therapeutic significance [20]. The compound 1,8-cineole has been shown to exhibit anti-inflammatory and analgesic properties [22] and is used in many oral health care products, such as mouthwashes and chewing gums [20]. Moreover, limonene and α-terpineol are common components in EEO. However, the concentrations of these individual compounds vary widely between species [23]. The oil composition also depends on the plant growing conditions, harvest time, extraction time, and temperature [21]. The consumption of EEO at low dosages is generally suggested to be safe for adults, however, if ingested at higher concentrations, EEO can cause systemic toxicity, especially in children. The common side effects of EEO poisoning in children encompass depression of consciousness, vomiting, and seizures. Severe poisoning in children has been reported after ingestion of 3 to 5 mL of pure EEO [24,25,26].

Recently, the immunomodulatory effects of EEO and its constituents have gained increased attention. Serafino et al. [34] investigated the impact of EEO on human monocyte-derived macrophages (MDMs) in vitro by confocal microscopy after the administration of fluorescent beads. The results showed drastically increased phagocytic activity in human MDMs during EEO treatment compared to those that were treated with LPS. In the untreated control groups, 13.7% of cells showed phagocytic activity with a mean of 11 phagocytosed beads per cell. LPS treatment for 6 h slightly increased the percentage of phagocytic cells to 18.26% and did not affect the number of phagocytosed beads, whereas treatment with 0.008% EEO increased the percentage to 27.1% with a mean of 24 phagocytosed beads per cell after 24 h of treatment. Additionally, pretreatment with EEO for 24 h before LPS challenge increased the phagocytic activity of human MDMs compared to that of LPS treatment alone. Active cell motility might contribute to the enhanced phagocytic ability, as MDMs treated with EEO demonstrated elongated lamellipodia and filopodia. A nonspecific effect of EEO on the phagocytic activity of MDMs was excluded by testing other oil preparations, which did not affect MDM phagocytic activity. Despite this stimulation, the LPS-induced production of proinflammatory cytokines was significantly reduced in cells that were pretreated with EEO. This effect was especially evident for the cytokines IL-4, IL-6, and TNFα. When nocodazole was added, the phagocytic activity of EEO-pretreated cells was inhibited, while the activity of LPS-stimulated cells was not influenced, suggesting that the EEO-mediated enhancement of phagocytosis was dependent on the microtubule network and was mediated through different mechanisms than LPS-induced phagocytosis, possibly involving different phagocytic receptors.

Furthermore, the study by Serafino et al. [34] revealed the impact of EEO on the phagocytic abilities of peripheral blood monocytes and granulocytes in immunocompetent and immunosuppressed rats after in vivo administration. In immunocompetent rats, EEO treatment significantly increased the percentage of circulating monocytes and simultaneously increased phagocytic activity and the expression of the CD44 receptor, which mediates adhesion to the endothelium, thus promoting extravasation [35]. In 5-fluorouracil-induced immunosuppressed rats, the administration of EEO led to a recovery of the percentage of circulating granulocytes and restored the phagocytic abilities of granulocytes and monocytes. Since 5-fluorouracil is commonly used in chemotherapy, these results also suggest a possible role of EEO in novel combination therapies to improve the treatment of cancer [34].

Yadav and Chandra [36] examined the effect of EEO and its main constituent 1,8-cineole on the phagocytic activity of lung alveolar macrophages. When cells were pretreated with 0.02% EEO 3 h before bacterial infection, an increase in the phagocytic activity of alveolar macrophages and intracellular pathogen clearance was observed. These results are consistent with those of Serafino et al. [34]. Additionally, pretreatment with EEO reduced the production of LPS-induced proinflammatory mediators, such as TNFα, IL-1β, IL-1α, and NO in lung alveolar macrophages. Furthermore, 1,8-cineole demonstrated a similar anti-inflammatory effect but only affected intracellular IL-1α, IL-1β, and IL-6. However, 1,8-cineole seems to exert a strong inhibitory effect on TNFα production in monocytes. In a previous study [37], 1,8-cineole pretreatment was shown to reduce TNFα and IL-1β production in human monocytes by 99% and 84%, respectively. Reduced IL-1β levels might result from the reduced expression of the nod-like receptor NLRP3 [36]. Unlike toll-like receptors, nod-like receptors are activated inside the cell and are part of the inflammasome, a multiprotein innate immune complex that is responsible for inflammatory responses. Active inflammasomes can lead to caspase 1-mediated activation of IL-1β [38]. This effect suggests a potential therapeutic impact of EEO on inflammatory diseases involving the inflammasome, such as type 2 diabetes [39], inflammatory bowel disease [40], and atherosclerosis [41].

In the study by Yadav and Chandra [36], EEO pretreatment also reduced mRNA expression of triggering receptor expressed on myeloid cells (TREM-1) [34]. TREM-1 induces the release of cytokines, such as IL-1β and TNFα, amplifying the inflammatory response [42]. Additionally, a reduction in LPS-induced phosphorylation of p38-mitogen-activated protein kinase (p38 MAPK) and nuclear factor kappa-light-chain-enhancer of activated B-cells (NFκB) was observed following EEO pretreatment [34]. P38 MAPK is associated with various chronic inflammatory diseases, such as rheumatoid arthritis and inflammatory bowel disease. Selective inhibition of p38 MAPK could lead to an important approach in the anti-inflammatory treatment of chronic diseases [43]. The decreased phosphorylation of NFκB and p38 MAPK and the mRNA expression of TREM-1 also correlate with the reduction in the levels of TNFα, IL-1β, IL-6, and NO. Moreover, 1,8-cineole increased the phosphorylation of NFκB in lung alveolar macrophages, suggesting that different constituents of EEO might be responsible for the reduction in NFκB activity [36]. In human monocytes, pretreatment with α-pinene was shown to inhibit the activity of NFκB [44], whereas 1,8-cineole demonstrated no effect on NFκB [45]. These results indicate that EEO exhibits different cell-type-specific effects on LPS-mediated inflammatory responses.

Hotta et al. [1] analyzed the impact of EOs on cyclooxygenase (COX-2) expression using bovine arterial endothelial cells and demonstrated that 0.01% EEO suppressed LPS-induced COX-2 promoter activity by 25%. Thyme and clove essential oils exhibited even stronger effects than those of EEO. While COX-1 is constitutively expressed in many cells and tissues, COX-2 is usually absent but is induced through numerous intra- and extracellular stimuli, including LPS and proinflammatory cytokines. Thus, COX-2 is a key mediator of inflammatory pathways [46]. Thyme and clove oils act as peroxisome proliferator-activated receptor (PPAR) agonists, thereby initiating a negative feedback loop that regulates COX-2 expression. EEO induced PPAR activation, but the effect was not significant. It is questionable whether single compounds in EEO exhibit stronger PPAR agonist effects than EEO itself to inhibit increased COX-2 expression. Further research is required to obtain a comprehensive outlook on this matter.

## 4. Clove Essential Oil

Along with *Eucalyptus*, clove (*Syzygium aromaticum*) belongs to the *Myrtaceae* family. Clove is an evergreen tree that is native to Indonesia and is mostly grown for its aromatic flowers. Clove essential oil (CEO) has been used in traditional medicine mainly as a pain reliever in dental care and in the treatment of burns and wounds. The main component in CEO is eugenol (Figure 2 and Table 1), a phenolic compound that contributes to the oil’s warm, spicy scent and exhibits many pharmacological properties [47]. The antioxidant, anti-inflammatory [48], antimicrobial [49], and analgesic [50] effects of eugenol have been previously documented. The oil is also composed of numerous other compounds, including eugenol acetate, β-caryophyllene, and α-humulene [51]. Generally, data on the cytotoxicity of EOs and their active cytotoxic components are limited. CEO has been shown to be highly toxic to human fibroblasts and endothelial cells at a concentration of 0.03%. This effect is mostly attributable to eugenol, however the cytotoxicity profile of CEO may be characterized by more than one constituent [52]. Additionally, CEO exhibits significant hepatotoxic effects. Ingestion of 10 mL CEO can cause hepatoxicity and renal dysfunction [53].

Many studies investigating the impact of CEO and its constituents on immune responses have resulted in conflicting findings. Carrasco et al. [54] examined the effect of CEO containing >98% eugenol on humoral and cell-mediated immune responses in mice in vivo. Daily oral administration of CEO (100, 200, and 400 mg/kg) for 7 days significantly increased the total white blood cell count in immunocompetent mice after immunization with sheep red blood cells (SRBCs) on day 0, and this effect was shown to be dose-dependent. In immunocompetent mice, CEO might stimulate immune responses by activating the hematopoietic system and increasing the number of circulating lymphocytes. Additionally, following cyclophosphamide-induced immunosuppression in mice, CEO (400 mg/kg) restored the total white blood cell count to the initial values after 7 days. While humoral immune responses were not significantly affected by CEO in immunocompetent mice, the production of circulating anti-SRBC antibodies drastically increased in cyclophosphamide-suppressed mice. A delayed-type hypersensitivity (DTH) assay was performed to identify the influence of CEO on cell-mediated immunity. After 24 h of antigen challenge, CEO-treated immunocompetent mice demonstrated a significant increase in foot-paw volume. CEO was effective in stimulating cell-mediated immunity in immunocompetent mice and in restoring the white blood cell counts and humoral immunity in immunosuppressed mice.

Conversely, in a similar study conducted by Halder et al. [55], 0.1 mL/kg CEO, which consisted of 87.34% eugenol, was administered daily for two weeks and led to a significant decrease in paw volume in rats. The authors suggest that CEO may indirectly reduce inflammation by inhibiting cell-mediated immune responses. Additionally, improvements in humoral primary and secondary immune responses were documented, which positively affected lymphocyte functions.

Islamuddin et al. [56] analyzed the immunomodulatory effect and therapeutic efficacy of eugenol emulsion (EE) in experimental visceral leishmaniasis (VL). After 10 days of EE treatment, enhanced cell-mediated immune responses, as identified by a significant augmentation in paw thickness and humoral immune responses in mice, were observed in vivo. EE treatment also led to an increase in IFNγ-secreting CD4^+^ and CD8^+^ splenic T cells, along with CD8^+^ central memory T lymphocytes. The generation of central memory cells was consistent with the upregulated expression of CD44 and CD62L, two adhesion molecules that are highly expressed on central memory T cells [56]. In peritoneal macrophages, EE significantly enhanced the expression of CD80 and CD86 [56]. These costimulatory molecules play pivotal roles in the activation of lymphocytes and the secretion of cytokines and are, therefore, critical to the initiation and maintenance of immune responses [57]. Additionally, the proliferation of antigen-stimulated splenocytes and lymphocytes and NO production were significantly augmented 10 days after EE treatment. EE also showed promising therapeutic potential in the treatment of VL by reducing hepatic and splenic parasitic burdens, as indicated by decreased spleen and liver weights. Furthermore, alterations in cytokine levels upon EE treatment were identified. While the production of the classic Th1 cytokines IFNγ and IL-2 increased, serum levels of cytokines released from Th2 cells, particularly IL-4 and IL-10, decreased. This finding provides further evidence for the suitability of EE for VL therapy, since IL-4 and IL-10 are associated with this disease [56].

Dibazar et al. [58] demonstrated both anti- and proinflammatory in vitro effects of CEO on LPS-stimulated mouse peritoneal macrophages. Significant suppression of NO and TNFα production and release were reported after 48 h of incubation. In previous studies, eugenol was shown to inhibit the expression of inducible nitric oxide synthase (iNOS), as well as the release of TNFα, in macrophages and might, therefore, be responsible for this effect [59,60]. Additionally, IL-6 production was stimulated in a particular experiment [58]. This effect might occur due to the reduced NO levels, since NO has been associated with the downregulation of LPS-induced IL-6 production in alveolar macrophages [61]. However, these findings are inconsistent with the results from Bachiega et al. [62], who demonstrated that CEO could inhibit the production of the proinflammatory cytokine IL-6 in murine macrophages. These contradictory results highlight the difficulties affiliated with evaluating the effects of CEO and its constituents, especially eugenol, on inflammation and on the overall immune response. As mentioned above, the bioactivity of EOs is influenced by many factors and varies with the chemical composition of the oil. The different geographical origin of the clove flower buds and extraction methods used in these two studies might be responsible for the obtained contradictory results. Further research with different doses of CEO and eugenol, as well as other target cell types, may provide a better understanding.

## 5. Tea Tree Essential Oil

*Melaleuca alternifolia*, commonly known as tea tree, is a member of the botanical family Myrtaceae and endemic to Australia. The essential oil from this tree has long been used in folk medicine as a topical medication to treat bruises and infected injuries. Tea tree essential oil (TTO) is composed of approximately 100 volatile compounds, the majority being terpene hydrocarbons. The main active component has been found to be terpinen-4-ol, a monoterpene with strong antimicrobial properties (Figure 2 and Table 1) [63]. TTO is toxic when ingested in higher concentrations and can cause skin irritation and allergic reactions in predisposed individuals. Due to their low body weight, children are more vulnerable to TTO poisoning. Ingestion of less than 10 mL pure TTO can cause central nervous system depression and ataxia in children. However, data on the toxicity profile of TTO are limited and the responsible components have not been identified [64].

Hart et al. [2] investigated the effect of TTO on the production of inflammatory mediators by LPS-activated human peripheral blood monocytes in vitro. The results indicated that some constituents of TTO appeared to be toxic to monocytes in culture, and the water-soluble components terpinen-4-ol, α-terpineol, and 1,8-cineole could suppress the production of inflammatory mediators in a nontoxic manner. Significantly reduced levels of TNFα, IL-1β, IL-8, IL-10, and prostaglandin E_2_ were identified after 40 h of incubation with the water-soluble components of TTO. In LPS-stimulated human macrophages, TTO demonstrated a similar inhibitory effect on IL-1β and IL-10 production, but TNFα levels remained unaltered [3]. These partially contradictory results suggest that the biological activity of TTO strongly depends on the cell type, as well as the composition and concentration of the oil. Both studies identified terpinen-4-ol as being primarily responsible for these effects, most likely by interfering with the NFκB, p38, or ERK/MAPK pathway.

Recently, the immunomodulatory effect of *Melaleuca alternifolia* concentrate (MAC) has been investigated. MAC represents a refined product derived from TTO that has very low concentrations of hydrophobic monoterpenes, but its major component is still terpinene-4-ol. Low et al. [65] examined the impact of MAC on protein expression in macrophage-like cell lines. MAC inhibits the phosphorylation of inhibitor of κB kinases (IκB), a mechanism that is necessary for the activation of NFκB and downstream target genes. Normally, IκB is bound to inactive NFκB, which is activated and released upon IκB phosphorylation [66]. This inhibition of NFκB translocation results in decreased levels of certain LPS-induced proinflammatory cytokines, as well as reduced iNOS expression and NO production [65]. Lee et al. [67] verified and extended these observations by demonstrating that MAC not only inhibited the phosphorylation of IκB but also increased its concentration in the cytosol. Additionally, the study revealed that MAC positively regulated heme oxygenase-1 (HO-1) expression by inducing the activation and translocation of NF-E2-related factor 2 (Nrf2). Since the Nrf2-HO-1 pathway is greatly involved in anti-inflammatory processes and inhibits the COX-2 and iNOS signaling pathways, these results suggest a promising role of TTO and its extracts in the treatment of inflammatory diseases.

Budhiraja et al. [68] also reported immunostimulatory effects of TTO. The study demonstrated that both TTO and terpinen-4-ol similarly induce the differentiation of immature myelocytes into active phagocytizing monocytes and increase the expression of CD11b, a receptor that is partially responsible for the phagocytosis of opsonized bacteria and fungi by leukocytes. While much research has been undertaken on the immunoinhibitory effects of TTO, little is known about its potential as an immunostimulatory agent.

## 6. Lavender Essential Oil

Lavender essential oil (LEO) is commonly obtained from *Lavandula angustifolia*, a species of flowering plants belonging to the *Lamiaceae* family that is widely cultivated in the Mediterranean region. This aromatic plant has a long history of use in traditional medicine as a natural remedy for various inflammatory disorders. LEO is steam-distilled from the flowers and leaves of the plant and contains high concentrations of the monoterpenoids linalool, linalyl-acetate, 1,8-cineole, camphor, and terpinen-4-ol (Figure 2 and Table 1). However, the oil composition strongly varies depending on the geographical origin of the plant, environmental factors and extraction parameters [69,70]. LEO is regarded as one of the mildest essential oils, however concerns about possible allergenic or irritant skin reactions are rising. LEO has been shown to be cytotoxic to human endothelial cells and fibroblasts at a concentration of 0.25%. The cytotoxicity is mainly attributable to the oil’s major constituent linalool and linalyl acetate [71]. Additionally, linalyl acetate has been previously shown to exhibit genotoxic effects on human lymphocytes [72].

Giovannini et al. [73] demonstrated a protective effect of LEO on human MDMs upon infection with *Staphylococcus aureus* in vitro, as indicated by a significant increase in phagocytic activity and a reduction in intracellular bacterial replication. The enhancement of bacterial clearance might occur due to the upregulated expression of genes involved in generating reactive oxygen species (CYBB and NCF4). These observations suggest that LEO could stimulate innate immune responses to bacteria. Additionally, LEO exerts regulatory effects on the inflammatory response by inhibiting the production of several bacterial-induced proinflammatory cytokines, including IL-1α, IL-1β, and IL-6, by macrophages. Similar to TTO, LEO is thought to exhibit its anti-inflammatory effects by inducing the expression of HO-1. The authors of this study could not identify the single constituent of LEO that is responsible for these immunomodulatory effects and instead suggested that the bioactivity may depend on the whole phytocomplex of the oil.

Chen et al. [74] investigated the impact of aromatherapy massage with 2% LEO on the stress level and immune functions of pregnant women. Twenty-four healthy pregnant women received 10 aromatherapy massages every other week for 20 weeks. Compared to the control group, who only received routine prenatal care, the intervention group demonstrated significantly elevated salivary IgA and reduced cortisol levels immediately after aromatherapy massage. Chen et al. also documented long-term effects on salivary IgA levels, which serve as indicators of immune function. In a similar study, aromatherapy massage with an oil blend of lavender, cypress, and sweet marjoram led to a drastic increase in peripheral blood lymphocytes in healthy subjects, presumably due to the documented increase in CD8^+^ T cells and CD16^+^ cells [75]. These results provide evidence that aromatherapy massage with LEO could significantly stimulate immune functions and diminish stress. Additionally, a significant impact of inhaled LEO was observed in the treatment of migraine headache in a placebo-controlled clinical trial [76]. Therefore, LEO might be an effective alternative for reducing symptoms caused by acute migraine.

Ueno-lio et al. [77] examined the potent anti-inflammatory effect of LEO on experimentally induced bronchial asthma in mice. Asthma is strongly driven by Th2 cells and their cytokines IL-4, IL-5, and IL-13, which induce eosinophilic inflammation in the airway [78]. Inhalation of LEO led to the suppression of allergic airway inflammation, as evidenced by reduced cell accumulation and mucus production. LEO decreased IL-5 and IL-13 production and inhibited eosinophilic infiltration. Additionally, mucus production was controlled through the downregulation of MUC5B, the gene encoding the major gel-forming mucin. The authors suggest that this downregulation might occur through the inhibition of NFκB activation induced by certain components of LEO, since NFκB has been found to stimulate MUC5B expression [79]. These observations reveal that LEO exhibits strong immunomodulatory effects and shows great potential as an alternative anti-inflammatory medicine for bronchial asthma.

## 7. Conclusions

This review highlights the growing interest in the immunomodulatory effects of plant-derived EOs and their main components. Protective attributes, such as antibacterial, antioxidative, or anti-inflammatory properties are already well described. However, the immunomodulatory effects of EOs have been considered only in a limited manner. Additionally, plant extracts induce few reported side effects compared to those of immunomodulatory pharmaceutical drugs. The current literature strengthens the potential of various EOs as suitable immunomodulatory alternative treatments for infectious or immune diseases. Furthermore, these compounds also provide good efficacy as preventive medicine, which promotes a general healthy lifestyle.

The results reviewed in this study revealed a significant reduction in relevant cytokines, such as IL-1α, IL-1β, IL-3, IL-4, IL-5, IL-6, IL-8, IL-10, IL-13, TNFα, NO, and IFNγ, which were measured mainly in monocytes and activated macrophages. Studies also showed both a reduction of proinflammatory cytokines by M1 activated macrophages, such as IL-1, IL6, NO, or TNFα, as well as the production of anti-inflammatory markers, such as IL-5, IL-13, expression of HO-1 (Nrf2-HO-1 pathway), increased production of CD8^+^, CD16^+^ cells, and IgA by activation of M2 macrophages. Additionally, selected EOs were identified to alter the NFκB and p38 MAPK pathways. The balance of these pathways and factors in pro- or anti-inflammatory functions remains critical in immunomodulation. Since macrophages play an important role in defense of microbial infections, tissue repair, and tumors, regulation via EOs might also stimulate results regarding these applications [80]. The main results of the reviewed articles are summarized in Table 2. Furthermore, EEO increased the phagocytic activities of macrophages and peripheral blood monocytes and enhanced bacterial clearance. EEO restored the number of circulating granulocytes and their phagocytic ability in immunosuppressed models. CEO additionally was shown to stimulate cell-mediated immunity in immunocompetent mice and restore total white blood cell count (WBC) and humoral immunity in immunosuppressed mice. Importantly, several studies presented in this review provided contradictory results. Here, multiple parameters, such as location and climatic conditions for cultivation, extraction procedures, applied concentration, general application, as well as the testing organism remain critical factors. The geographical origin, environment, and extraction parameters highly affect the final extract composition. Variations in the main composition, but also changes in minor substances, can lead to great variations, representing a challenging task for commercial exploitation and exploration [81,82]. Chemometric approaches might reliably predict the bioactivity of multicomponent substances in the future for better characterization of EOs and an improved comparability within studies [8,9]. Due to hydrophobic properties and viscosity, the application of EOs to in vitro cell culture remains very challenging, which might also contribute to contradictory results across the reviewed studies. The applied dose and application (oral, inhalation, skin) also influences the EO efficacy. Finally, the selected in vivo organism or in vitro cell type strongly influences EO performance, rendering a precise prediction highly difficult.

The effects of EOs have mainly been examined at the cellular level, including in monocytes, macrophages, and Th cells, in the context of the molecular impacts on cytokines or immunoglobulins. Only a few studies addressed possible genetic regulations and mechanisms [83]. Further research should focus on the genetic regulatory pathways involved. Additionally, most of the analyzed data were derived from in vitro cell culture and in vivo mouse/rat experiments. In fact, there is a huge gap between in vitro studies and in vivo or clinical trials. Besides aromatherapy clinical trials, few data are available describing the immunomodulatory effects of EOs in humans. The U.S. Food and Drug Administration (FDA) considers many EOs as “Generally Recognized As Safe” (GRAS), therefore very little attempt has been made to investigate toxic effects on the body. Nevertheless, multiple studies revealed the toxic effects of EOs, clearly showing the need for appropriate testing systems for identifying their modes of action in the metabolism. Recently, even low concentrations of selected EOs showed toxic potential regarding respiratory disorders, mucous membrane irritation, acute toxicity, and organ toxicity [84,85]. This holds true also for the selected essential oils in this review. EEO was shown to cause acute poisoning symptoms, CEO revealed hepatocytic effects, and TTO and LEO exhibit skin irritation and allergic reactions. There is also evidence of cytotoxic effects in selected cell types. Hence, there is a great demand for comprehensive toxicity testing prior to applications in food, feed, and pharmaceutical products. Alternative systems, such as 3D-cell culture, organs-on-chip, or in vivo models like *Caenorhabditis elegans* and *Drosophila melanogaster*, might lead to novel insights and predictive approaches in this emerging topic.

Especially for determining effective and critical toxicological values as well as acute toxicity, more research should be conducted in alternative model organisms. This fact represents a key point for future therapeutic applications. In this regards, multifactor approaches will be necessary in order to avoid over- or underestimation of the toxicological properties resulting in unjustified restrictions or safety attestation.

However, EOs have been successfully applied as feed additives to broiler chickens or weaned piglets and have clearly shown immunostimulatory effects, while no biotoxicity was observed. Additionally, more clinical trials focusing on immunomodulation are needed, since in vitro cell culture and in vivo experiments showed strong evidence in the context of the immunomodulatory properties of selected EOs.

## Figures and Tables

**Figure 1 biomolecules-10-01139-f001:**
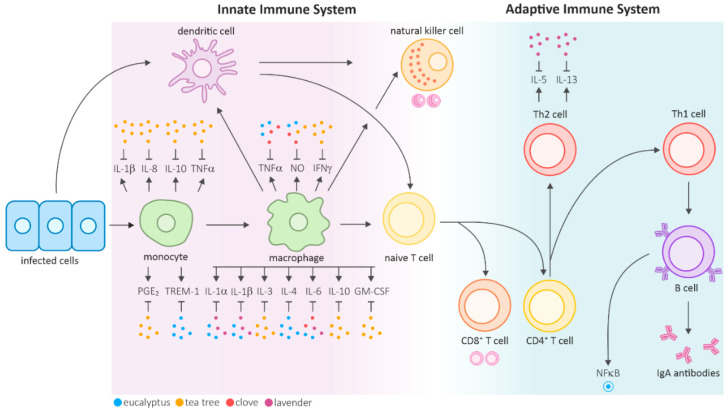
Overview of the mammalian innate and adaptive immune systems and the regulatory effects of selected essential oils on various cytokines shown as dots (adapted from [11]). Antigen-presenting cells (APCs), such as dendritic cells or macrophages, are recruited to the infection site. The released cytokines further activate natural killer cells and lead to the maturation of T cells. The reviewed studies identified multiple cytokines that were downregulated by eucalyptus, tea tree, clove, and lavender EOs. Specific interleukins and other factors are also expressed in other cell types. Most reviewed research articles focused on monocytes and activated macrophages. Hence, only those effects are highlighted in the figure.

**Figure 2 biomolecules-10-01139-f002:**
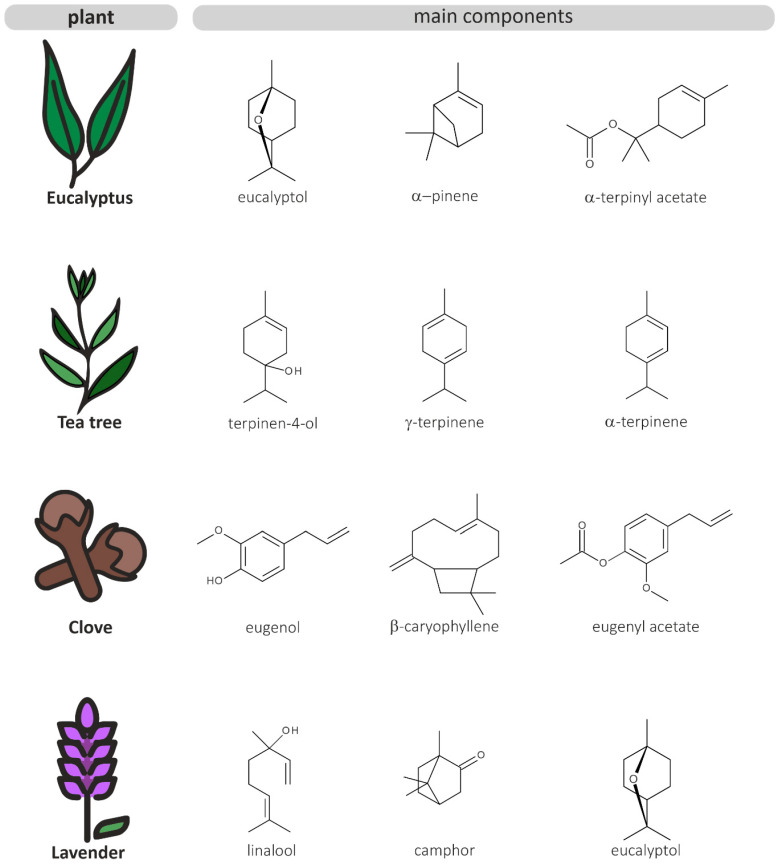
Chemical structures of the main components of the four selected EOs.

**Table 1 biomolecules-10-01139-t001:** Main components of selected essential oils.

Eucalyptus [23,27]		Clove [28,29]		Tea Tree [30,31]		Lavender [32,33]	
*Compound*	*Content*	*Compound*	*Content*	*Compound*	*Content*	*Compound*	*Content*
eucalyptol	50–80%	eugenol	70–76%	terpinen-4-ol	30–48%	linalool	28–45%
α-pinene	2–26%	β-caryophyllene	10–17%	γ-terpinene	10–28%	camphor	3–12%
α-terpinyl acetate	2–5%	α-humulene	2%	α-terpinene	5–13%	eucalyptol	2–10%
α-terpineol	2–3%	eugenyl acetate	1–12%	α-terpineol	1–8%	terpinen-4-ol	2–7%
trans-pinocarveol	1–8%	α-cubebene	1–2%	*p*-cymene	1–8%	β-caryophyllene	1–6%
globulol	1–6%	α-copaene	1–2%	α-pinene	1–6%	borneol	1–10%
limonene	1–4%	nerolidol	0–1%	limonene	1–3%	limonene	0–3%
pinocarvone	1–4%	farnesol	<1%	sabinene	0–4%	α-pinene	0–2%
γ-terpinene	1–2%	methyl chavicol	<1%	δ-cadinene	0–3%	β-pinene	0–2%
*p*-cymene	1–7%	caryophyllene oxide	<1%	eucalyptol	0–15%	myrcene	0–2%

**Table 2 biomolecules-10-01139-t002:** Overview of the immunomodulatory effects and concentrations of selected essential oils.

Essential Oil	Effects on Immune Functions	Concentrations
*Eucalyptus*	Increased the phagocytic activities of macrophages and peripheral blood monocytes and enhanced bacterial clearance [34,36]; restored the number of circulating granulocytes and their phagocytic ability in immunosuppressed models [34]; inhibited the production of IL-1α, IL-1β, IL-4, IL-6, TNFα, and NO [34,36]; attenuated the activation of p38 MAPK, NFκB, and TREM-1 [36]; suppressed COX-2 promoter activity by 25% [1]	EEO 0.008 and 0.016% [*v*/*v*] (in vitro), EEO 12 mg/kg/day for 15 days (in vivo) [34]; EEO and 1,8-cineole 0.02% [*v*/*v*] (in vitro) [36]; EEO 0.01% (in vitro) [1]
*Clove*	Many contradictory results; stimulated cell-mediated immunity in immunocompetent mice and restored WBC count and humoral immunity in immunosuppressed mice [54]; inhibited cell-mediated responses and improved humoral immune responses in immunocompetent rats [55]; suppressed NO and TNFα production by macrophages [58]; stimulated [58] and inhibited [62] IL-6 production; enhanced cell-mediated and humoral immune responses in experimental VL [56]; suppressed COX-2 promoter activity by 40% [1]	CEO (<98% eugenol) 100, 200, 400 mg/kg/day for 7 days (in vivo) [54]; CEO (87.34% eugenol) 0.1 mL/kg/day (in vivo) [55]; ethanolic CEO extract (74% eugenol), aqueous CEO extract (43% eugenol) 0.001–1000 µg/mL (in vitro) [58]; clove extract 100 µg/well, eugenol extract 50 and 100 µg/well (in vitro) [62]; eugenol emulsion 25, 50, and 75 mg/kg/day for 10 days (in vivo) [56]; CEO 0.01% (in vitro) [1]
*Tea tree*	Stimulated the differentiation of immature myelocytes into active phagocytizing monocytes and increased CD11b receptor expression [68]; suppressed the production of TNFα, IL-1β, IL-8, IL-10, and prostaglandin E_2_ by blood peripheral monocytes [2]; MAC reduced the production of NO and proinflammatory cytokines, inhibited NFκB activation and induced HO-1 expression [65,67]	TTO and terpinen-4-ol 20-90 µmol/L (in vitro) [68]; water soluble components of TTO at 0.125% (42% terpinen-4-ol, 3% α-terpineol and 2% 1,8-cineole) (in vitro) [2]; MAC (60–64% terpinen-4-ol, 8–14% p-cymene) 0.004–0.016% [*v*/*v*] (in vitro) [65]; MAC (60% terpinen-4-ol) 0.01–0.5% (in vitro) [67]
*Lavender*	Increased the phagocytic activity of macrophages and reduced intracellular bacterial replication and the production of IL-1α, IL-1β, and IL-6 [73]; attenuated IL-5 and IL-13 secretion and inhibited eosinophilic infiltration and mucus production in mouse asthma models [77]; aromatherapy massage increased IgA levels [74] and the number of CD8^+^ and CD16^+^ cells [75]	LEO (39% linalool, 11.97% camphor, 10.54% eucalyptol) dilution of 1:50,000 for 10^6^ cells (in vitro) [73]; LEO (31.78% linalyl acetate, 25.56% linalool) 20 µL on 10 × 10 filter paper (in vivo) [77]; LEO 2% (aromatherapy clinical trial) [74]; essential oil blend of lavender (36.31% linalool, 34.05% linalyl acetate), cypress (61.85% β-pinene, 15.2% 3-carene), and sweet marjoram (21.26% terpinen-4-ol, 13.46% γ-terpinene) (aromatherapy clinical trial) [75]

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
