# Peer review of "Immunomodulatory Activities of Selected Essential Oils"

_biomolecules, 2020, doi:10.3390/biom10081139_

Round 1

Reviewer 1 Report

The manuscript is very interesting and brings a summary of knowledge about immunomodulatory activities of essential oils. Half of the abstract is very general and includes long-known things. This part can be improved. Conclusion is very general also and needs to be improved with data about eucalyptus, clove, tea tree  and lavender essential oils. In conclusion manuscript is very interesting and needs just to improve abstract and conclusion.

Author Response

Thank you for acknowledging the quality of our study and for helping to improve our manuscript.

Half of the abstract is very general and includes long-known things. This part can be improved. Conclusion is very general also and needs to be improved with data about eucalyptus, clove, tea tree and lavender essential oils.

We changed the abstract and conclusion as requested.

Reviewer 2 Report

The review reports data about the possible immunomodulatory activity of FOUR essential oils.

For this, in my opinion, the title should be changed to "Immunomodulatory activities of SOME essential oils".

Some points to be addressed.

  1. Lines 31-32: Authors reported that the bioactivity of the essential oils is due to their major component(s). It is correct, but recent studies based on chemometric analyses revealed the role also of minor components or the synergism among consitutents in determining the bioactivity.
  2. Authors should cite the problem of the variability in the composition of the essential oils du to endogenous and hexogenous factors and the consequent effects on biological activities.
  3. Most of researches are related to in vitro experiments and there is a enormous gap between in vitro and in vivo and human studies and clinical applications. Authors reported a sentence about this, but in my opinion this matter should be discussed.
  4. Even it a review, information about methods and materials must be added: how Authors gathered their information, the source of this information, etc.

Author Response

Thank you for your valuable comments to improve our manuscript. We adapted the manuscript as following:

According to your suggestion the title as changed to “Immunomodulatory Activities of selected Essential Oils”.

Lines 31-32: Authors reported that the bioactivity of the essential oils is due to their major component(s). It is correct, but recent studies based on chemometric analyses revealed the role also of minor components or the synergism among constituents in determining the bioactivity.

Changed as requested.

Authors should cite the problem of the variability in the composition of the essential oils du to endogenous and exogenous factors and the consequent effects on biological activities.

The problem is now addressed in the introduction as well as in the conclusion.

Most of researches are related to in vitro experiments and there is a enormous gap between in vitro and in vivo and human studies and clinical applications. Authors reported a sentence about this, but in my opinion this matter should be discussed.

We improved the conclusion by adding additional information.

Even it a review, information about methods and materials must be added: how Authors gathered their information, the source of this information, etc.

We added the requested information in the “2. Methods” section.

Reviewer 3 Report

Sandner and colleagues provide an overview of immune activities of some essential oils with emphasis in the anti-inflammatory effects.

Sandner and colleagues provide an overview of immune activities of some essential oils with emphasis in the anti-inflammatory effects.

Below are some suggestions and questions.

1. The authors should define immunomodulation, immunoregulation and immunostimulation. These concepts are confused throughout the text.

2. A figure illustrating the plants and main components of the selected oils (main chemical structures) could be interesting and impact the text.

3. Why did authors select these essential oils? Please add this information in the text.

4. How the authors did the literature research (criteria, search engines etc)?

5. Toxicity of the selected essential oils should be discussed in more detail.

6. Discussing the Peterfalvi review (Much more than a pleasant scent: a review on essential oils supporting the immune system, Molecules 2019, 24(24), 4530), could help to identify more cellular effects of essential oils.

7. In the table 1 consider including the main chemical constituents and doses, and delete toxicity data.

8. A phenotype change in M1 / M2 macrophages appears to occur in several of the experiments discussed in the text. Comments on this might lead to fresh insights into the essential oils immune activities.

9. A more detailed discussion about inconsistent data cited in many parts of this review might also lead to fresh insights into the essential oils immune activities.

10. The chemical studies of the essential oils using omics approaches could be discussed by the authors as well as the methods to solve inconsistent results in the biological experiments.

Author Response

Thank you for your valuable time and feedback helping to improve our manuscript. We changed the manuscript as following:

  1. The authors should define immunomodulation, immunoregulation and immunostimulation. These concepts are confused throughout the text.

The terms are now defined in line 89-93.

  1. A figure illustrating the plants and main components of the selected oils (main chemical structures) could be interesting and impact the text.

Figure 2 illustrating the plants and their main chemical structures was added, as suggested.

  1. Why did authors select these essential oils? Please add this information in the text.

Selection of common EOs was based on literature research which is now clarified in the added “2. Methods” section.

  1. How the authors did the literature research (criteria, search engines etc)?

As for question 3, the information was added to “2. Methods” section.

  1. Toxicity of the selected essential oils should be discussed in more detail.

Toxicity is added to the essential oil section. Furthermore, toxicity is discussed in detail in the conclusion.

  1. Discussing the Peterfalvi review (Much more than a pleasant scent: a review on essential oils supporting the immune system, Molecules 2019, 24(24), 4530), could help to identify more cellular effects of essential oils.

The review was added in the conclusion section.

  1. In the table 1 consider including the main chemical constituents and doses, and delete toxicity data.

According to your suggestion, we removed toxicity data and replaced them with tested compounds and doses.

  1. A phenotype change in M1 / M2 macrophages appears to occur in several of the experiments discussed in the text. Comments on this might lead to fresh insights into the essential oils immune activities.

We added your input in the conclusion for further discussion within this regard.

  1. A more detailed discussion about inconsistent data cited in many parts of this review might also lead to fresh insights into the essential oils immune activities.

We improved the discussion about inconsistent data and toxicity.

  1. The chemical studies of the essential oils using omics approaches could be discussed by the authors as well as the methods to solve inconsistent results in the biological experiments.

We added your suggestion in the introduction and also discussed it in the conclusion.

Round 2

Reviewer 1 Report

I don't have comments to change, all previous comments were accepted.

Reviewer 3 Report

The authors have responded well to the previous comments